# ESG Rating and Northbound Capital Shareholding Preferences: Evidence from China

**Guochao Wan [1,2,*] and Ahmad Yahya Dawod [1]**

1   International College of Digital Innovation, Chiang Mai University, Chiang Mai 50200, Thailand;
ahmadyahyadawod.a@cmu.ac.th
2   School of Management, Chengdu University of Information Technology, Chengdu 610225, China
*   Correspondence: guochao_wan@cmu.ac.th

**Abstract:** In the context of achieving carbon peak and carbon neutrality goals and the opening of a capital market in China, an emerging country, the relationship between an ESG rating and northbound capital shareholding preferences (NCSP) is a topic worthy of discussion. In this research, we selected CSI 300-listed companies from 2015 to 2020 as the research object and examined the influence and mechanism of the ESG rating on the NCSP. Our findings showed that the ESG rating is significantly correlated with NCSP, that the ESG rating can dramatically enhance corporate accounting conservatism, and that accounting conservatism has a partial mediating effect between an ESG rating and the NCSP. Furthermore, we noticed that the positive effect of ESG ratings on NCSP among non-state (non-SOE) corporations is more pronounced. The most prominent of the three perspectives of ESG ratings was governance (G). We found that the ESG rating had a stronger impact on the NCSP during the post-COVID-19 period than in the pre-COVID-19 period. In this paper, based on the perspective of accounting conservatism, we enrich the study of ESG ratings in the capital market, provide empirical evidence for the theoretical study of NCSP, and offer a reference for the optimization of the ESG concept and its positioning in corporations. In future studies, expanding the sample range may lead to different interesting findings.

**Keywords:** ESG rating; accounting conservatism; northbound capital shareholding preferences; mediating effect





## 1. Introduction

Environmental, social, and governance (ESG) is the value of sustainable development, which aims at a harmonious coexistence between humans and nature. With the progress made towards sustainable development, it has become the consensus of all sectors of society that enterprises need to broaden their social responsibility to a wider range. According to the Global Sustainable Investment Alliance (GSIA), the management scale of global ESG assets increased from USD 13 trillion in early 2012 to USD 35 trillion by early 2020, representing an increase of 169.2%. Currently, international interest in responsible investment is also increasing rapidly, and as of February 2022, more than 4700 institutions worldwide had joined the UN Principles for Responsible Investment (PRI), including 90 from China. The ESG rating is an important tool for assessing ESG performance and can help all participants in the capital market to judge the ESG performance of companies. The rising interest of investors in sustainable corporate performance has contributed to the rapid development of ESG ratings and by 2020, more than 600 providers had offered ESG ratings around the world, including about 20 in China. To achieve the goals of 'peak carbon' and 'carbon neutrality' in China, whether ESG ratings can provide investors with information related to an investment value has become a common concern for the capital market and other stakeholders.

Since China implemented its 'Shanghai–Hong Kong Stock Connect Program', which allows Hong Kong investors to trade A shares within a specified range through security

companies, the inflow of Hong Kong and international capital into the stock market through the 'Shanghai–Hong Kong Stock Connect Program' and the 'Shenzhen–Hong Kong Stock Connect Program' has been called 'Northbound Capital'. In recent years, following the initial liberalization of China's capital market, northbound capital has continued to be dynamic, and its shareholding ratio in listed companies has gradually increased [1]. In 2021, the total turnover of northbound capital exceeded RMB 25 trillion, with a net purchase of RMB 432.1 billion, indicating that northbound capital has gradually cemented its importance in China's capital market. Foreign institutional investors represented by northbound capital have more advanced analytical tools, richer globalization experience [2], more information advantages [3], higher levels of social trust [4], and good corporate governance structures [5], are more willing to invest in a transparent information environment [6], and can help to improve the performance of invested companies [7]. ESG ratings convey non-financial information about a company to the public, and investors can assess the ESG performance of a company comprehensively. According to the *Global Institutional Investor Survey 2021* published by Morgan Stanley Capital International (MSCI), 52% of the 200 institutional investors investigated claimed to have adopted ESG investment strategies, and 73% planned to increase their scale in ESG investments by the end of 2021. This shows the importance that institutional investors attach to ESG. The existing literature indicates that responsible institutional investors tend to be more patient with high-ESG firms [8]. Meanwhile, mutual funds with good sustainability ratings can obtain inflows and poor ratings bring about negative flows to mutual funds [9]. Moreover, the increased focus in socially responsible institutions on ESG may influence their stock return patterns [10] and it would be interesting to study an ESG rating and its impact on stock returns from an institutional investment perspective; however, whether ESG ratings can influence northbound capital shareholding preferences (NCSP) in China, an emerging market, is a question that has not been answered in any literature thus far.

Accounting conservatism is a fundamental principle in business accounting and aims at measuring the quality of accounting information [11]. The existing literature shows that accounting conservatism can improve investment efficiency [12] and limit management's misuse of cash [13]. Under unfavorable macroeconomic conditions and financial constraints, management prioritizes accounting conservatism instead of engaging in corporate social responsibility (CSR); however, Shen et al. showed that CSR can be effectively used to promote accounting conservatism in China [14]. Although CSR is closely related to accounting conservatism, few researchers have investigated the overall role of ESG, rather than just its role in relation to social responsibility. The question of whether ESG ratings influence accounting conservatism and what role accounting conservatism can play in the relationship between ESG ratings and NCSP is worthy of attention.

This paper investigates the effect of the ESG rating on NCSP and the mediating role of accounting conservatism based on the ESG rating data of a sample of listed companies in the CSI 300-listed companies published by SynTao Green Finance during the period 2015–2020. The results of the regression indicate that there is a significant positive relationship between an ESG rating and the NCSP. In addition, our study finds that accounting conservatism has a partial mediating effect on the above relationship. We also chose to use the Heckman two-stage approach for robustness testing, which guarantees the reliability of the results, by changing the core independent variables and lagging one period to check the robustness. Further study found that the G dimension and non-state-owned (non-SOE) companies have the largest positive effect on NCSP. Additionally, ESG ratings had a stronger impact on NCSP during the post-COVID-19 period than in the pre-COVID-19 period.

Compared with the existing literature, the innovations of our study are as follows. First, the existing literature focus on the impact of one aspect of environment, society, or governance on foreign investors' shareholding preferences, whereas we investigated for the first time the overall ESG ratings of foreign investors' shareholding preferences in emerging markets. This not only provides recent evidence from China on the overall ESG shareholding preference effect, but also extends the literature on ESG ratings. Second,

compared with most of the literature, which adopts QFII in order to evaluate foreign institutional investors, we chose to use northbound capital as the measure of foreign investors' shareholding preferences and to test the relationship between northbound capital and the ESG rating. This further enriches the research in the field of foreign investors' shareholding preferences and provides a reference for decision making by investors and relevant government regulators. Third, we clarified the mechanism of the influence of an ESG rating on NCSP and provided a preliminary exploration of the mediating effect of accounting conservatism. Moreover, the degree of influence of the ESG rating on NCSP was explored from the perspectives of different ownership properties and the different dimensions of E, S and G. We also considered the impact of COVID-19 on NCSP. The multi-perspective analysis provided an important addition to the study of the economic consequences of ESG ratings and provided a reference for the study of ESG ratings in the emerging market. The findings obtained will help companies to maintain a good reputation and establish a perfect investor protection mechanism.

Our paper contains a relevant literature review and develops the research hypothesis in Section 2. Additionally, Section 3 presents the relevant data and methodology. We provide details of our empirical results and the robustness tests used in Section 4 and describe our extensibility analysis in Section 5. The last section provides our conclusions and the implications of this research.

## 2. Literature Review and Research Hypothesis

### 2.1. ESG Rating and Northbound Capital Shareholding Preferences

Capital market liberalization would enhance the quality of firm disclosure and reduce the cost of financing for firms [15], which could help to increase corporate value. According to the signal theory, high-quality ESG performance is a positive signal to foreign investors and enhances the attractiveness of companies. In the context of sustainability theory, companies that perform well in ESG demonstrate outstanding strengths in resource utilization, social relationships, and corporate governance, all of which help them maintain stable earnings and achieve more sustainable development during volatile market conditions [16]. Institutional investors tend to prefer listed companies with a good financial status and high profitability [17]. Moreover, companies with a high ESG rating have a complete corporate governance mechanism, a robust regulatory system, and incentive management regulations, which reduce their agency costs and improve their financial performance [18]; this effect has been found to be more significant for large companies [19]. In addition, the non-financial information covered by ESG is beneficial for guiding the future production and operation of companies; this can help foreign investors to understand the company's situation more comprehensively and enhance their confidence in its long-term development. Meanwhile, positive factors such as environmental protection and social responsibility enhance the likelihood of a company receiving government support and more bank loans, thereby helping companies to obtain better financing. The public focus on corporate social responsibility and green environmental protection is consistently increasing, and investors are more inclined to choose to invest in reputable companies to reduce the risk of capital recovery [20]. Companies can maintain a sustainable competitive advantage by cultivating a good reputation. When corporate scandals relating to environmental pollution, waste of resources, and problems with social responsibility occur, companies may experience high levels of consumer resentment, damaging their reputation. In contrast, a good corporate social responsibility record can enhance a company's reputation and attract investors' attention [21]. For example, although the revenue of ERKE in the past was lower than that of many other sports brands, the company directly donated RMB 50 million in resources when a flood struck in Henan Province. This behavior led to intense discussion among internet users, who began to purchase many ERKE products, resulting in the brand value reaching second place in the industry. In summary, based on signal theory, the non-financial information signaled to the public through ESG provides data with which overseas institutional

investors can evaluate the value of a corporation objectively and minimize their investment risk. Based on the above analysis, we propose H1.

**Hypothesis 1 (H1).** *ESG rating is positively correlated with NCSP, which means that the higher the ESG rating is, the more significant the NCSP will be.*

## 2.2. ESG Rating and Accounting Conservatism

We expect that companies with high ESG ratings will have more pronounced accounting conservatism in three areas: environment, social responsibility, and corporate governance. Regarding the environment, based on sustainability theory, companies should ensure the sustainability of environmental benefits in their production and operation. Being environmentally responsible increases shareholder value as well as the value of non-financial stakeholders [22]. The government has reduced tax to encourage companies to control the emission of pollutants, and this tax reduction policy could compensate for the initial expenditure of companies on environmental investment while bringing immediate and rapid increases in profits, encouraging them to increase their accounting conservatism. Green credit policies could force companies that cause high levels of pollution to obtain loans with higher interest rates from financial institutions [23], which could lead to them reducing their accounting conservatism. An environmental commitment is an important tool for companies to build their social capital and achieve a good environmental performance [24]; it could encourage companies to build harmonious relationships with local communities to avoid issues such as disputes. Additionally, environmentally friendly companies could receive community support, which could ease tensions between companies and regulators and reduce the cost of violations, contributing to accounting conservatism.

In terms of social responsibility, based on stakeholder theory, companies should comprehensively measure the needs of each stakeholder and in addition to focusing on financial performance, companies should also concentrate on the social benefits they bring, with good social responsibility being likely to be viewed positively by stakeholders. For consumers, having positive social responsibility could convey a good image to consumers [25]. In the context of market information asymmetry, consumers prefer to choose products from companies that have a brand premium, and these companies will fulfill their social responsibilities well. When negative events occur, the public is more tolerant of companies with high levels of social responsibility, and these companies suffer less reputation damage [26]. Conversely, the possibility that negative reports of social responsibility could trigger the involvement of government agencies exists, and this could pose a potential risk to the company and, thus, reduce its accounting conservatism. Moreover, the fulfillment of social responsibility could receive some government support, such as tax incentives and financial subsidies. For market participants, the value-driven theory suggests that CSR can improve the transparency of accounting information and inhibit management's surplus management behavior [27]. Shareholders and creditors are more willing to invest in socially responsible companies, and, thus, these companies are more likely to obtain financing. Banks and other financial institutions also judge the credibility of companies based on their level of social responsibility. For example, in terms of employees, companies with good social responsibility are more likely to gain high-quality research and development employees, who will promote the upgrading of the company's industrial structure and product quality, reduce the company's operational risk, improve its financial performance, and, thus, enhance the accounting conservatism of the company. Moreover, companies with socially responsible behavior tend to avoid lucrative insider trading [28], surplus management [29], corporate fraud [30], and corruption [31].

Regarding corporate governance, the agency theory suggests that the operation of modern companies is based on the perspective of profit maximization and that management is more inclined to focus on short-term interests, leading to 'short-term behavior' in which it will make decisions that deviate from the maximization of shareholders' wealth [32]. By improving their governance, companies can mitigate the risk of management manipulating

surpluses while ignoring the interests of other stakeholders. Appropriate management compensation incentives and equity incentives are used to prevent short-term interest behavior, reduce agency costs, and improve accounting conservatism. Companies with strong governance have been shown to have higher levels of accounting conservatism [33]. Meanwhile, good governance can also lead to the creation of a more reasonable and standardized employee promotion mechanism, which greatly improves employee enthusiasm and corporate performance, which are beneficial to enhancing accounting conservatism. This paper then proposes Hypothesis 2.

**Hypothesis 2 (H2).** *ESG rating is positively correlated with accounting conservatism, meaning that the higher the ESG rating is, the stronger the accounting conservatism will be.*

*2.3. The Mediation Role of Accounting Conservatism*

Due to the existence of geographical, linguistic, and cultural limitations, foreign investors prefer to invest in companies with high-quality information disclosure [34]. Accounting information can provide external stakeholders with information about the financial status and operational performance of a company and provide pricing and governance functions [35], which is an important basis for investment decisions. Accounting conservatism, which is the main characteristic of accounting information quality, can provide information to investors in an incremental fashion, thus alleviating the information disadvantage of foreign investors.

First, accounting conservatism can not only discourage overinvestment by companies but also mitigate the problem of underinvestment. According to the principle of accounting conservatism, executives disclose unfavorable information that they do not intend to disclose for reasons such as enhancing their reputation and obtaining protection from regulatory penalties [36]. Accounting conservatism also leads to the recognition of losses in investment projects in a timely manner during their tenure, leading them to abstain from investing in projects with a negative net present value, discouraging over-investment behavior from management, and reducing the investment risk of foreign investors. Meanwhile, managers can suffer from overconfidence and tend to overestimate the corporate share prices. When companies fail to raise sufficient funds through debt, managers can tend to worry that equity financing would harm the interests of the original shareholders and, therefore, abandon the inflow of external funds due to self-interest, leading to underinvestment. In addition, accounting conservatism can effectively alleviate the information asymmetry existing between firms and foreign investors and lower the cost of outside funding, which helps firms obtain new funds to invest in promising investment projects to mitigate the problem of underinvestment. Second, accounting conservatism increases the cost of surplus manipulation by management, reduces the incentive for surplus management, and satisfies the needs for foreign investors, thus leading to obtaining a more complete understanding of the company's financial position. This can help a company's future cash flows and operating conditions to be analyzed, thus reducing the risk they might face and decreasing their share price volatility [37]. Companies with good accounting conservatism with higher levels of revenues and assets are also more likely to have reserves set aside for special situations, and the implementation of prudential principles helps to prevent and mitigate the uncertainty of foreign investors. Finally, one of the most important ways for Chinese companies to obtain financing is from banks and other financial institutions. The higher the conservatism of accounting information is, the faster the recognition of bad news is delivered and the lower the delay in the recognition of good news. In this situation, banks would know the potential risks of a business, which would lead to them increasing their level of trust and lowering their loan rates [38]. This indicates that companies with stronger accounting conservatism are more likely to attract foreign investors, thus influencing the NCSP.

In conclusion, favorable ESG performance can enhance accounting conservatism, which can improve NCSP; therefore, accounting conservatism can be regarded as an

intermediate variable between the ESG rating and the NCSP, which means that an ESG rating not only has a direct impact on NCSP but that it can also indirectly influence the NCSP through accounting conservatism. Based on the above analysis, we propose Hypothesis 3 and Hypothesis 4.

**Hypothesis 3 (H3).** *Accounting conservatism has a positive relationship with NCSP.*

**Hypothesis 4 (H4).** *Accounting conservatism has a mediating effect between the ESG performance of companies and NCSP.*

The four hypotheses are summarized in Figure 1.

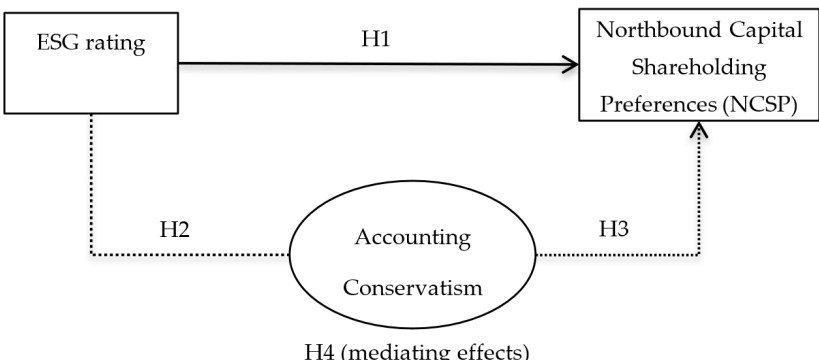

**Figure 1.** Proposed conceptual framework.

## 3. Data and Methodology

### 3.1. Data and Sample Selection

This paper uses CSI 300-listed companies during the 2015–2020 period as a research sample for empirical analysis. The ESG rating is used for the quantitative evaluation of enterprise ESG performance [39] and has been widely applied in the research of Chinese ESG issues [40,41]. To reasonably evaluate the enterprise ESG rating, third-party rating agencies conduct comprehensive evaluations of the three aspects of ESG to obtain the environmental performance, social responsibility performance, and corporate governance level, thus, providing an enterprise ESG rating. In this study, the ESG rating data were sourced from SynTao Green Finance, which has gradually disclosed Chinese listed companies' ESG ratings information and data since 2015 and covers a relatively wide number of companies, while other financial data were obtained from the China Stock Market and Accounting Research (CSMAR) database. All data were annual unbalanced data. To ensure the accuracy and reliability of the data, the following criteria were used to select the sample data: (1) exclude listed companies in financial and insurance industries, (2) exclude all kinds of samples that are ST or *ST, (3) exclude companies whose ESG rating information is absent, and (4) exclude enterprises that lack data. After the exclusion of the above data, 1943 firm-year observations were included in the final sample. Meanwhile, all continuous variables were winsorized at the head and tail 1% positions.

### 3.2. Definition of Variables

#### 3.2.1. Dependent Variable

Northbound capital shareholding preference (NCSP) was the dependent variable in this study. In our paper, the dummy variables used for foreign capital shareholding preferences were measured after the implementation of the trading system of the 'Shanghai–Shenzhen–Hong Kong Stock Connect' and the companies that were the subject of the 'Shanghai Stock Connect' and the 'Shenzhen Stock Connect'. The Shanghai–Hong Kong Stock Connect was launched on 17 November 2014, and the Shenzhen–Hong Kong Stock Connect was launched on 5 December 2016. As the time of the opening of 'Shenzhen–Hong Kong

Stock Connect' was during 2015, we used the opening time as the policy implementation point (Post), meaning that after this, the policy implementation (Post) takes the value of 1, with it otherwise being 0. From the perspective of northbound capital, if the top ten shareholders of the sample companies included Hong Kong Securities Clearing Company (HKSCC) Limited, (Treat) takes the value of 1; otherwise, it takes the value of 0. In practice, northbound capital is known as 'hot money' or 'smart money' and has been regarded as the market investment 'vane'; many times, it has successfully bought at the bottom of the A-share market in China. The 'Certain Provisions of the Interoperability Mechanism between the Mainland and Hong Kong Stock Markets' clearly state that HKSCC Limited, as the holder nominated by Hong Kong investors to perform shareholder rights on their behalf, can best represent the NCSP. Since currently listed companies only disclose the information of the top ten shareholders, when the top ten outstanding shareholders of the sample companies included HKSCC Limited, (Treat) takes the value of 1, and 0 if otherwise. In this paper, we study the preferences of northbound capital shareholding; thus, we chose the interaction term of double difference (Post_Treat) to measure the NCSP and study the relationship between the ESG rating and NCSP of the CSI 300-listed companies in China. We also used the dummy variable of HKSCC limited being in the top ten shareholders of the company to check the robustness.

### 3.2.2. Independent Variable

In our paper, the ESG rating data were selected as the independent variable. As in Deng and Cheng [41] and Broadstock et al. [42], the ESG rating data used in our study came from the third-party performance agency SynTao Green Finance and were obtained by collecting the ESG-related information of Chinese companies with quantified and evaluated information, before the ESG information was finally converted into scores and ratings. According to the SynTao Green Finance ESG Rating Methodology, the ESG score is composed of the ESG management score and ESG risk score. ESG ratings are based on ESG scores. In our paper, we selected the ESG rating data of CSI 300-listed companies from SynTao Green Finance for the period 2015–2020. Table 1 provides the indicators for the ESG rating framework. These indicators contain four grades and 10 levels (A+, A, A−, B+, B, B−, C+, C, C−, D). Through assignment in turn, we assigned A+ to 1, A to 2, A− to 3, B+ to 4, B to 5, B− to 6, C+ to 7, C to 8, C− to 9, and D to 10.

**Table 1.** SynTao Green Finance dataset rating system tiers.

| Tier 1 | Tier 2 | Tier 3 |
|---|---|---|
| E (Environmental) | E1: Environmental Management | Environmental management system certification, water conservation objectives, green products (services), etc. |
| | E2: Environmental Disclosure | Energy consumption and conservation, waste gas emission and reduction |
| | E3: Environmental Controversies | Negative incidents regarding water pollution/air pollution/solid waste pollution |
| S (Social) | S1: Employees | Freedom of association, anti-discrimination |
| | S2: Supply Chain | Responsible supply chain management |
| | S3: Customer Management | Customer information confidentiality, etc. |
| | S4: Community | Community communication |
| | S5: Product | Fair trade products, genetically modified food |
| | S6: Philanthropy | Enterprise foundation, donation |
| | S7: Social Controversies | Negative incidents regarding employees/clients/etc. |

**Table 1.** *Cont.*

| Tier 1 | Tier 2 | Tier 3 |
|---|---|---|
| | G1: Business Ethics | Whistle-blowing policy, overseas tax payment |
| G (Governance) | G2: Corporate Governance | Information disclosure, independence of the supervisory board, executive compensation, diversity of the board of directors, etc. |
| | G3: Governance Controversies | Business ethics, corporate governance negative events |

Note: drawn by the author according to the system from SynTao in China.

### 3.2.3. Mediator Variable

Accounting conservatism is the mediating variable in this study. Accounting conservatism, an important part of accounting information quality, requires significantly more positive news than negative news when making disclosures [14]. We chose to use the extended model C_Score [43], based on the modified Basu (1997) model, to measure the accounting conservatism. The calculation was as follows:

$$\text{EPS}_{i,t}/\text{P}_{i,t-1} = \mu_0 + \mu_1\text{RET}_{i,t} + \mu_2\text{D}_{i,t} + \mu_3\text{D}_{i,t} \times \text{RET}_{i,t} + \varepsilon_{i,t} \tag{1}$$

$$\text{G\_Score}_{i,t} = \mu_1 = \alpha_1 + \alpha_2\text{SIZE}_{i,t} + \alpha_3\text{MB}_{i,t} + \alpha_4\text{LEV}_{i,t} \tag{2}$$

$$\text{C\_Score}_{i,t} = \mu_3 = \beta_1 + \beta_2\text{SIZE}_{i,t} + \beta_3\text{MB}_{i,t} + \beta_4\text{LEV}_{i,t} \tag{3}$$

$$\begin{aligned}\text{EPS}_{i,t}/\text{P}_{i,t-1} = {} & \mu_0 + \mu_2\text{D}_{i,t} + (\alpha_1 + \alpha_2\text{SIZE}_{i,t} + \alpha_3\text{MB}_{i,t} + \alpha_4\text{LEV}_{i,t}) \times \text{RET}_{i,t} \\ & + (\beta_1 + \beta_2\text{SIZE}_{i,t} + \beta_3\text{MB}_{i,t} + \beta_4\text{LEV}_{i,t}) \times \text{D}_{i,t} \times \text{RET}_{i,t} + \varepsilon_{i,t}\end{aligned} \tag{4}$$

$\text{EPS}_{i,t}$ shows the earnings per share of the company i in year t, $\text{P}_{i,t-1}$ shows the stock price of the company i at the end of year t − 1, and $\text{RET}_{it}$ indicates the stock yield of the company i in year t. Di takes a value of 1 if $\text{RET}_{it} < 0$ and 0 otherwise. $\text{SIZE}_i$ indicates the size of the company i, which is obtained from ln(total assets). $\text{LEV}_i$ shows the gearing ratio of company i, and $\text{MB}_i$ shows the ratio of bookings to the market of company i. The G_Score and C_Score models are shown in Equations (2) and (3), respectively. $\mu_1$ represents the speed of the company's response to good news and ($\mu_2 + \mu_3$) represents the speed of the company's response to bad news; therefore, $\mu_3$ reflects the incremental speed of the company's response to a bad report compared to its response to a good report, which reflects the level of accounting conservatism of the firm. Equations (2) and (3) are then substituted into Equation (1) to obtain Equation (4) for regression to find the coefficients $\beta_1$, $\beta_2$, $\beta_3$, and $\beta_4$. Finally, the parameter values are substituted back into Equation (3) to find the annual C_Score value of the company, and the larger the value is, the stronger the accounting conservatism of the firm will be.

### 3.2.4. Control Variables

According to the prior literature on this subject, we selected the following control variables in our paper. Firm size (Size): companies of different sizes will have different market shares, risk resistances, and levels of market attention, which affect how attractive they are to northbound capital. Gearing ratio (LEV): the higher the gearing ratio is, the higher the possibility that the company will experience a financial crisis and the higher the uncertainty faced by investors in the future will be. Earnings per share (EPS): investors can assess the profitability and management of a company through their earnings per share, which has a guiding effect on investors. Firm market valuation (Tobin's Q): the higher the market value of a firm is, the better the ability of the company to create wealth will be, which can reduce the problem of internal financing constraints and lead to companies facing a lower financial risk. Additionally, institutional investors are more likely to prefer well-financed companies. Earnings management (EM): earnings management can cause information asymmetry between investors and managers, resulting in higher transaction and agency costs and reducing the quality of accounting information. Whether to hire

the Big 4 (Big4) for an auditor: large firms have a higher level of practitioner service and can provide effective guaranteed mechanisms and signaling mechanisms. These reduce the probability of a firm being penalized and make it easier to attract offshore investors. Relevant definitions of the variables are shown in Table 2.

**Table 2.** Variable definitions.

| Type of Variable | Variable Name | Symbol | Explanation of Variable |
|---|---|---|---|
| Dependent variable | Northbound capital shareholding preferences | Post_Treat | The interaction term of the variable 'Post' and the variable 'Treat' |
| Independent variable | ESG rating | ESG | According to SynTao Green Finance ESG ratings from low to high, the value is 1~6 |
| Mediator variable | Accounting conservatism | C_Score | Incremental sensitivity of accounting surplus to 'bad news' over 'good news' |
| Control variables | Firm Size | Size | Natural logarithm of total number of company employees |
| | Gearing ratio | LEV | The ratio of total liabilities to total assets |
| | Earnings per share | EPS | The ratio of profit after tax to total equity |
| | Firm market valuation | Tobin's Q | (market capitalization of equity + book value of total liabilities)/book value of total assets |
| | Earnings management | EM | Total assets/total equity |
| | Whether to hire 'the Big 4' auditors | Big4 | External audit by a 'Big 4' firm in the current year is assigned a value of 1, with the opposite being assigned a value of 0 |

Note: Table 2 reports the symbol and definitions for all the variables. All data were sourced from the CSMAR database, except for the ESG rating data, which were collected from the third-party performance agency, SynTao Green Finance, a leading consultancy specializing in providing ESG data and green finance advice in China.

### 3.3. Model

To investigate whether the accounting conservatism of the listed companies in a sample can affect the NCSP under the influence of the ESG rating and based on the mediation effect test procedures proposed by Wen et al. [44], we established the following models.

To test H1 (the relationship between the ESG rating and NCSP of CSI 300-listed companies in China), Model 1 was established as follows

:

$$\text{Post\_Treat}_{i,t} = \alpha_0 + \alpha_1 \text{ESG}_{i,t} + \alpha_2 \text{Size}_{i,t} + \alpha_3 \text{LEV}_{i,t} + \alpha_4 \text{EPS}_{i,t} + \alpha_5 \text{Tobin's Q}_{i,t} + \alpha_6 \text{EM}_{i,t} + \alpha_7 \text{Big4}_{i,t} + \varepsilon_{i,t}$$

To test H2 (the relationship between the ESG rating and accounting conservatism of CSI 300-listed companies in China), Model 2 was established as follows:

$$\text{C\_Score}_{i,t} = \alpha_0 + \alpha_1 \text{ESG}_{i,t} + \alpha_2 \text{Size}_{i,t} + \alpha_3 \text{LEV}_{i,t} + \alpha_4 \text{EPS}_{i,t} + \alpha_5 \text{Tobin'Q}_{i,t} + \alpha_6 \text{EM}_{i,t} + \alpha_7 \text{Big4}_{i,t} + \varepsilon_{i,t}$$

To test H3 (the relationship between the accounting conservatism and NCSP of CSI 300-listed companies in China), Model 3 was established as follows:

$$\text{Post\_Treat}_{i,t} = \alpha_0 + \alpha_1 \text{C\_Score}_{i,t} + \alpha_2 \text{Size}_{i,t} + \alpha_3 \text{LEV}_{i,t} + \alpha_4 \text{EPS}_{i,t} + \alpha_5 \text{Tobin'sQ}_{i,t} + \alpha_6 \text{EM}_{i,t} + \alpha_7 \text{Big4}_{i,t} + \varepsilon_{i,t}$$

To test H4 (accounting conservatism plays a mediating effect between the ESG rating and NCSP of CSI 300-listed companies in China), Model 4 was established as follows:

$$\text{Post\_Treat}_{i,t} = \alpha_0 + \alpha_1 \text{ESG}_{i,t} + \alpha_2 \text{C\_Score}_{i,t} + \alpha_3 \text{Size}_{i,t} + \alpha_4 \text{LEV}_{i,t} + \alpha_5 \text{EPS}_{i,t} + \alpha_6 \text{Tobin'Q}_{i,t} + \alpha_7 EM_{i,t} + \alpha_8 \text{Big4}_{i,t} + \varepsilon_{i,t}$$

where $\alpha$ = intercept and $\varepsilon$ = error items.

Table 1 shows a summary of the definitions of the variables used in this paper.

## 4. Empirical Results

### 4.1. Descriptive Statistics

Table 3 shows the results of the descriptive statistics for all variables used in the models. From Table 2, it can be seen that the average Post_Treat level was 0.472, indicating that nearly half of the sample companies received investment from northbound capital. The highest ESG rating was six while the lowest was one, indicating that there were differences in the ESG performance among the sample companies. The mean value was 3.044, which means that more than half of the sample companies focused on investing in ESG. The standard deviation (SD) of the ESG rating was 1.037, indicating that the ESG rating of the sample listed firms varied greatly. The mean value of accounting conservatism (C_Score) was negative, suggesting that the accounting conservatism of the sample companies needs to be improved. In terms of the control variables, the mean firm size (Size) was about 10. The mean of the gearing ratio (LEV) was approximately 50%, indicating that most companies could afford high gearing ratios. The earnings per share (EPS) of the best companies was 8.173 and the worst was −1.657. The variance of the earnings management (EM) was 1.167, which indicates that the level of earnings management varied significantly among the companies. The mean value for the sample companies hiring from the Big 4 (Big 4) was 0.223, indicating that most companies did not hire Big 4 firms for auditing.

**Table 3.** Descriptive statistics.

| Variables | Obs | Mean | SD | Min | Max |
|---|---|---|---|---|---|
| Post_Treat | 1943 | 0.472 | 0.499 | 0.000 | 1.000 |
| ESG | 1943 | 3.044 | 1.037 | 1.000 | 6.000 |
| C_Score | 1943 | −0.026 | 0.074 | −0.232 | 0.199 |
| Size | 1943 | 9.298 | 1.303 | 5.576 | 12.905 |
| LEV | 1943 | 0.486 | 0.193 | 0.049 | 0.868 |
| EPS | 1943 | 0.820 | 1.085 | −1.657 | 8.173 |
| Tobin's Q | 1943 | 2.287 | 2.066 | 0.784 | 14.197 |
| EM | 1943 | 2.327 | 1.167 | 1.052 | 7.578 |
| Big4 | 1943 | 0.223 | 0.417 | 0.000 | 1.000 |

Note: Table 3 shows the results of the descriptive statistics for the variables defined in Table 2.

### 4.2. Correlation Analysis

As the multicollinearity problem between the variables usually affects the reliability of the regression results, we chose to use Pearson's correlation coefficient analysis. This helped us to increase the credibility of the research hypothesis and should have provided more credible conclusions. The matrix of the correlation coefficients between the variables is shown in Table 4. The NCSP (Post_Treat) was found to be positively and significantly correlated with ESG, accounting conservatism (C_Score), Size, EPS, Tobin's Q, and Big4; However, there was a negatively significant correlation between the ESG rating and accounting conservatism, which was inconsistent with hypothesis two. In addition, the absolute values of the correlation coefficients of the independent variables were almost all less than 0.5, implying A low multicollinearity of these variables. The last column of Table 3 shows the results of the VIF test, where the largest variable value was 6.07 and all the VIF values were less than 10; Therefore, the independent variables did not cause multicollinearity problems.

**Table 4.** Analysis of the correlation between variables.

| | Variables | Post_Treat | ESG | C_Score | Size | LEV | EPS | Tobin's Q | EM | Big4 | VIF |
|---|---|---|---|---|---|---|---|---|---|---|---|
| 1 | Post_Treat | 1 | | | | | | | | | 2.35 |
| 2 | ESG | 0.181 *** | 1 | | | | | | | | 1.08 |
| 3 | C_Score | 0.053 ** | 0.064 *** | 1 | | | | | | | 1.52 |
| 4 | Size | 0.172 *** | 0.179 *** | −0.266 *** | 1 | | | | | | 1.70 |
| 5 | LEV | −0.010 | 0.053 ** | 0.294 *** | 0.401 *** | 1 | | | | | 6.07 |
| 6 | EPS | 0.274 *** | 0.047 ** | −0.209 *** | 0.158 *** | −0.086 *** | 1 | | | | 1.14 |
| 7 | Tobin's Q | 0.097 *** | −0.028 | 0.049 ** | −0.329 *** | −0.456 *** | 0.192 *** | 1 | | | 1.44 |
| 8 | EM | −0.027 | −0.017 | 0.207 *** | 0.301 *** | 0.872 *** | −0.034 | −0.351 *** | 1 | | 4.52 |
| 9 | Big4 | 0.092 *** | 0.232 *** | −0.264 *** | 0.408 *** | 0.156 *** | 0.093 *** | −0.168 *** | 0.111 *** | 1 | 1.29 |

Note: *** and ** indicate 1%, and 5% significance levels, respectively.

### *4.3. Multiple Regression Analysis*

### 4.3.1. ESG Rating and Northbound Capital Shareholding Preferences

As Post_Treat is a binary variable, we adopted a panel Logit model for the analysis. Referring to Chen (2015) [45], we applied the Hausman test to discriminate between the FE model and the RE model. The results of the Hausman test ($\chi^2 = 97.47$, $p < 0.01$) favored the use of the FE model. Column (1) of Table 5 shows the results obtained from an FE panel logit estimation with all variables included. The coefficient of the ESG rating in the regression of NCSP was positively significant at 1%, implying that the ESG rating had a positive relationship with the NCSP and that hypothesis 1 is valid. Column (2) of Table 5 indicates the result of the FE model when only significant variables were retained. The results were similar to those of column (1). Column (3) presents the results obtained for the RE estimation, which were qualitatively similar to those of the FE model in column (1), although the Hausman test favors the FE model. We chose to use the FE model for analysis in the following text.

**Table 5.** Regression results of ESG rating on NCSP.

| | (1) | (2) | (3) |
|---|---|---|---|
| **Variables** | **FE** | **FE** | **RE** |
| | **Post_Treat** | **Post_Treat** | **Post_Treat** |
| ESG | 0.930 *** | 1.063 *** | 0.473 *** |
| | (6.74) | (8.35) | (6.48) |
| Size | 3.455 *** | | 0.357 *** |
| | (8.70) | | (4.93) |
| LEV | 2.819 | | 0.807 |
| | (1.64) | | (1.01) |
| EPS | 0.501 *** | | 0.699 *** |
| | (3.65) | | (8.32) |
| Tobin's Q | 0.036 | | 0.133 *** |
| | (0.55) | | (3.42) |
| EM | −0.167 | | −0.147 |
| | (−0.58) | | (−1.18) |
| Big4 | −0.298 | | −0.159 |
| | (-0.45) | | (−0.82) |
| Constant | | | −5.757 *** |
| | | | (−8.04) |
| Observations | 1513 | 1513 | 1943 |

Note: *** indicate 1% significance levels.

### 4.3.2. ESG Rating, Accounting Conservatism, and NCSP

In column (1) of Table 6, it can be seen that the regression coefficient of the ESG rating and accounting conservatism was 0.010 at the 1% confidence level (the corresponding *t*-value was 5.21), indicating that the ESG rating can significantly enhance accounting conservatism, thus validating hypothesis 2. Column (2) of Table 6 indicates that the

regression coefficient between accounting conservatism and the NCSP was 25.991 at the 1% significance level (the corresponding *t*-value was 12.50), indicating that accounting conservatism can improve NCSP, which supports hypothesis 3. Meanwhile, on the basis of H1, H2, and H3, the ESG rating was still significantly positive at the 1% level and the regression coefficient was 0.792 (the corresponding *t*-value was 5.14) according to column (3) in Table 6, meaning that there is a partial mediating effect and that hypothesis 4 is validated. This means that accounting conservatism plays a mediating effect between the ESG performance of companies and the NCSP.

**Table 6.** Regression results obtained for accounting conservatism.

| Variables | (1) | (2) | (3) |
|---|---|---|---|
| | C_Score | Post_Treat | Post_Treat |
| ESG | 0.010 *** | | 0.792 *** |
| | (5.21) | | (5.14) |
| C_Score | | 25.991 *** | 24.965 *** |
| | | (12.50) | (11.90) |
| Size | −0.012 ** | 4.107 *** | 3.748 *** |
| | (−2.26) | (9.72) | (8.77) |
| LEV | 0.367 *** | −5.330 *** | −5.035 ** |
| | (10.97) | (−2.65) | (−2.40) |
| EPS | −0.003 | 0.593 *** | 0.587 *** |
| | (−1.56) | (3.84) | (3.74) |
| Tobin's Q | −0.007 *** | 0.277 *** | 0.267 *** |
| | (−5.89) | (3.83) | (3.65) |
| EM | −0.007 | −0.011 | 0.060 |
| | (−1.37) | (−0.03) | (0.16) |
| Big4 | −0.040 *** | 0.299 | 0.221 |
| | (−4.02) | (0.41) | (0.28) |
| Constant | −0.079 * | | |
| | (−1.66) | | |
| R-squared | 0.242 | | |
| Observations | 1943 | 1513 | 1513 |

Note: ***, **, and * indicate 1%, 5%, and 10% significance levels, respectively.

*4.4. Robustness Tests*

4.4.1. Robustness Checks Using Heckman Two-Stage Method

We used the Heckman two-stage method to control the potentially endogenous enhanced robustness between the ESG ratings and NCSP. Referring to Li et al. (2021) [46], in the first step, the ESG rating was converted into a dummy variable (Dum_ESG) according to the annual average of the ESG rating. When the ESG rating was higher than the annual average, Dum_ESG was 1, and it was 0 otherwise. Through the Probit model, the Inverse Mills Ratio (IMR) was calculated. In the second step, we input the IMR value into model 1 as a control variable to participate in the regression. The regression results of the Heckman two-stage test are reported in columns (1) and (2) of Table 7, showing that the results in Tables 5 and 6 remained constant.

4.4.2. Alternative Dependent Variable

We changed the core dependent variable to ensure our results. For the measurement of NCSP, we chose to use the shareholding of HKSCC Limited (Post_Treat1), as shown in Table 8. The regression results are shown in Table 9. A regression analysis was conducted again with the new measure of the NCSP, and the results of the regression coefficients among the main variables showed little change, confirming that the regression results were robust.

**Table 7.** Robustness tests performed using the Heckman two-stage method.

| Variables | Heckman Two-Stage Test | |
| --- | --- | --- |
| | **(1)** | **(2)** |
| | **Dum_ESG** | **Post_Treat** |
| ESG | | 0.931 *** |
| | | (6.75) |
| IMR | | −15.925 |
| | | (−0.27) |
| Size | −1.676 *** | 30.031 |
| | (−5.04) | (0.31) |
| LEV | −1.868 | 32.394 |
| | (−0.88) | (0.30) |
| EPS | −0.614 *** | 10.236 |
| | (−3.35) | (0.28) |
| Tobin's Q | 0.359 *** | −5.652 |
| | (4.64) | (−0.27) |
| EM | 0.071 | −1.284 |
| | (0.22) | (−0.31) |
| Big4 | 0.535 | −8.794 |
| | (0.87) | (−0.28) |
| Observations | 1015 | 1513 |

Note: *** indicate 1% significance levels.

**Table 8.** Ratios of the top ten shareholders included in HKSCC Limited.

| Shareholding Ratios | Values |
| --- | --- |
| Holding = 0 | 0 |
| 0% < holding ≤ 1% | 1 |
| 1% < holding ≤ 5% | 2 |
| 5% < holding ≤ 10% | 3 |
| 10% < holding ≤ 50% | 4 |
| Holding > 50% | 5 |

**Table 9.** Alternative measure of the core dependent variable.

| Variables | (1) | (2) | (3) | (4) |
| --- | --- | --- | --- | --- |
| | **Post_Treat1** | **C_Score** | **Post_Treat1** | **Post_Treat1** |
| ESG | 0.338 *** | 0.010 *** | | 0.262 *** |
| | (5.17) | (5.21) | | (4.10) |
| C_Score | | | 7.982 *** | 7.537 *** |
| | | | (11.63) | (10.76) |
| Size | 0.555 *** | −0.012 ** | 0.694 *** | 0.645 *** |
| | (5.48) | (−2.26) | (7.06) | (6.89) |
| LEV | 0.425 | 0.367 *** | −2.430 *** | −2.344 *** |
| | (0.55) | (10.97) | (−3.08) | (−3.05) |
| EPS | 0.174 *** | −0.003 | 0.198 *** | 0.198 *** |
| | (2.81) | (−1.56) | (3.24) | (3.34) |
| Tobin's Q | 0.010 | −0.007 *** | 0.069 *** | 0.062 *** |
| | (0.45) | (−5.89) | (2.93) | (2.62) |
| EM | 0.083 | −0.007 | 0.130 | 0.137 |
| | (0.79) | (−1.37) | (1.39) | (1.50) |
| Big4 | −0.189 | −0.040 *** | 0.119 | 0.112 |
| | (−0.70) | (−4.02) | (0.40) | (0.41) |
| Constant | −5.560 *** | −0.079 * | −4.567 *** | −4.961 *** |
| | (−5.72) | (−1.66) | (−5.09) | (−5.67) |
| R-squared | 0.075 | 0.242 | 0.134 | 0.149 |
| Observations | 1943 | 1943 | 1943 | 1943 |

Note: ***, **, and * indicate 1%, 5%, and 10% significance levels, respectively.

### 4.4.3. Independent Variable Lagged by One Year

Considering the possible robustness of the ESG rating and NCSP, the meaningful ESG rating information is published at the end of the year, and there is a delay in the process of providing feedback to foreign investors. Therefore, in this research, we re-ran the regression of the ESG rating with a lag of one period; the results obtained are shown in Table 10. From comparing with Table 5, we can see that the results were generally consistent with the empirical results obtained before the replacement, which shows the robustness of our results.

**Table 10.** Independent variable lagged by one year.

| Variables | (1) | (2) | (3) | (4) |
|---|---|---|---|---|
| | Post_Treat | C_Score | Post_Treat | Post_Treat |
| L_ESG | 0.893 *** | 0.009 *** | | 0.762 *** |
| | (5.48) | (3.62) | | (4.27) |
| C_Score | | | 25.991 *** | 21.283 *** |
| | | | (12.50) | (9.65) |
| Size | 3.631 *** | 0.002 | 4.107 *** | 3.979 *** |
| | (7.02) | (0.29) | (9.72) | (7.25) |
| LEV | 5.059 ** | 0.389 *** | −5.330 *** | −1.445 |
| | (2.28) | (8.40) | (−2.65) | (−0.56) |
| EPS | 0.367 ** | −0.004 | 0.593 *** | 0.412 ** |
| | (2.39) | (−1.58) | (3.84) | (2.37) |
| Tobin'Q | 0.154 * | −0.009 *** | 0.277 *** | 0.331 *** |
| | (1.83) | (−6.41) | (3.83) | (3.91) |
| EM | −0.378 | −0.014 | −0.011 | −0.274 |
| | (−0.97) | (−1.64) | (−0.03) | (−0.57) |
| Big4 | 0.625 | −0.033 *** | 0.299 | 1.308 |
| | (0.68) | (−2.71) | (0.41) | (1.08) |
| Constant | | −0.195 *** | | |
| | | (−3.39) | | |
| R-squared | | 0.205 | | |
| Observations | 1171 | 1572 | 1513 | 1171 |

Note: ***, **, and * indicate 1%, 5%, and 10% significance levels, respectively.

## 5. Extensibility Analysis

### 5.1. Impact Analysis of Different Dimensions of E, S, and G Rating on NCSP

ESG ratings may differ from individual assessments of E, S, and G. The above analysis suggests that firms with high ESG ratings would be favored by northbound capital; however, it is also worthwhile considering further which sub-dimension has the greatest effect. Based on the scores obtained for each part of E, S, and G, we assigned the scores as shown in Table 11 and then conducted a regression analysis with E, S, and G as the core independent variables. The results are shown in column (1) of Table 12. Overall, environmental protection, social responsibility, and corporate governance were found to be positively associated with NCSP, with corporate governance being the most prominent of these. For further analysis, we input the scores obtained for E, S, and G into the regression at the same time; the results indicated that the coefficients of E, S, and G were 0.507, 0.187, and 0.564, respectively. We can also see that corporate governance had the greatest effects, which means that the G dimension is the most important for NCSP. This may be because China is currently in the initial stage of ESG development, and high-quality corporate governance can be reflected in time to enhance corporate effectiveness to achieve sustainable development; however, stakeholders may be concerned that if companies do not have sufficient internal resources to finance their growth and choose to invest in corporate environmental protection and social responsibility activities, they may miss out on profitable projects and become increasingly unstable. The time spans of the E, S, and G breakdowns and the positive benefits of conservation and social responsibility may take longer to appear; thus, there may be some element of uncertainty in the conclusions drawn at this time.

**Table 11.** Assignment of different dimensions of E, S, and G.

| Scores | Values |
|:---:|:---:|
| X ≤ 40 | 1 |
| 40 ≤ X < 45 | 2 |
| 45 ≤ X < 50 | 3 |
| 50 ≤ X < 55 | 4 |
| 55 ≤ X < 60 | 5 |
| X ≥ 60 | 6 |

**Table 12.** Results based on different dimensions, different property rights, and period relative to COVID-19.

| Variables | (1) | (2) | (3) | (4) | (5) |
|:---:|:---:|:---:|:---:|:---:|:---:|
| | Post_Treat | Post_Treat | Post_Treat | Post_Treat | Post_Treat |
| | **Different Dimensions** | **SOEs** | **Non-SOEs** | **Post-COVID-19** | **Pre-COVID-19** |
| E | 0.507 *** | | | | |
| | (4.86) | | | | |
| S | 0.187 * | | | | |
| | (1.57) | | | | |
| G | 0.564 *** | | | | |
| | (5.35) | | | | |
| ESG | | 0.761 *** | 1.121 *** | 0.342 *** | 0.251 *** |
| | | (0.170) | (0.252) | (0.153) | (0.056) |
| Size | 3.343 *** | 2.316 *** | 4.979 *** | 0.199 | 0.256 *** |
| | (8.47) | (0.560) | (0.612) | (0.142) | (0.056) |
| LEV | 2.636 | 1.457 | 2.601 | 1.705 | 0.625 |
| | (1.49) | (2.569) | (2.715) | (1.658) | (0.670) |
| EPS | 0.482 *** | 1.034 *** | 0.090 | 0.942 *** | 0.656 *** |
| | (3.49) | (0.219) | (0.177) | (0.253) | (0.088) |
| Tobin's Q | 0.057 | −0.019 | 0.131 | 0.671 *** | 0.082 ** |
| | (0.85) | (0.118) | (0.093) | (0.239) | (0.032) |
| EM | −0.172 | −0.177 | 0.233 | −0.059 | −0.180 * |
| | (−0.56) | (0.373) | (0.581) | (0.258) | (0.109) |
| Big4 | −0.375 | −1.195 | 1.771 | −0.581 | 0.055 |
| | (−0.54) | (0.837) | (1.308) | (0.396) | (0.145) |
| Constant | | | | −3.852 *** | −4.138 *** |
| | | | | (1.336) | (0.541) |
| Observations | 1513 | 798 | 690 | 350 | 1593 |

Note: ***, **, and * indicate 1%, 5%, and 10% significance levels, respectively.

## *5.2. Heterogeneity Analysis Based on the Nature of Different Property Rights*

Considering the different property rights of companies, their resources, policy preferences, and business purposes will be different. State-owned enterprises (SOEs) have natural political relations with governments [24,47]; therefore, we further classified the companies in this sample into SOE and non-SOE companies according to the nature of their ownership. This helped us to investigate the effect of the ESG rating on the NCSP under the differing natures of ownership. We defined the SOEs as 1; otherwise, we defined them as 0. Column (2) and (3) in Table 12 show that the coefficients for the SOE and non-SOE companies were 0.0761 and 1.121, respectively, which were significant at the 10% confidence level. The coefficients of the non-SOE companies were larger than those of the SOE companies. This proves that the NCSP in non-SOE companies are better than that in SOE companies. Since SOE companies' ESG practices focus on responding to government requirements and fulfilling policies rather than obtaining economic benefits, investors believe that they should perform well in ESG practices. Non-SOE companies, on the other hand, tend to practice ESG primarily to satisfy the demands of stakeholders

who can bring about economic benefits, aiming to enhance their reputation and attract consumers to create economic returns. This leads to an improved ESG performance having a greater benefit in enhancing the NCSP. For another, benefits may not necessarily be gained from improving the ESG performance because SOE companies have a higher likelihood of receiving government support and credit concessions from financial institutions. The difficulties experienced by non-SOE companies in obtaining these benefits leads to the greater need for these companies to improve their ESG performances in order to earn customers' trust and support.

### 5.3. Impact Analysis of ESG Rating on NCSP around the COVID-19 Period

Table 12 also presents the results obtained for the different effects of ESG ratings on NCSP during the period of the COVID-19 pandemic. As the final year of our data was 2020, there would be an effect from the impact of COVID-19 on ESG investors' preferences. We defined 2020 as post-COVID-19, and the other years as pre-COVID-19. In column (4) and column (5), based on the regression coefficient, which indicates the different effects during the COVID-19 pandemic period, we found that the regression coefficient for the post-COVID-19 period (0.342) was higher than that for the pre-COVID-19 period (0.251). This result proves that the ESG rating is having a greater impact on the NCSP in the post-COVID-19 period than that in the pre-COVID-19 period, which may be related to the confidence of investors being improved by better ESG performances by firms. These results imply that high ESG ratings have a positive impact on NCSP during the post-COVID-19 period. This result is similar to that found by Broadstock et al. [42].

### 6. Conclusions

In this paper, we used 2015–2020 CSI 300-listed companies in China as our research object and analyzed the influence mechanism of ESG rating on NCSP by constructing a relationship model of ESG rating–accounting conservatism–northbound capital shareholding preferences. This study shows that the ESG rating has a positive effect on the NCSP. We found that an improvement in ESG performance not only has positive effects for companies and enhances the confidence of foreign institutional investors, but it also eases agency costs and financing constraints, thus, improving profitability. This is conducive to building a good corporate reputation and reducing the risk for foreign institutional investors. Moreover, we found that accounting conservatism plays a partial mediating role between the ESG rating and NCSP. A good company ESG performance can lead to a higher NCSP by improving the accounting conservatism. Moreover, our results remained unchanged after using the Heckman two-step method, changing the core independent variables, and introducing a lag time of one period. Furthermore, we also found that the ESG rating has a more significant effect on the NCSP among non-SOE companies and that the G dimension has the largest positive effect on this relationship. Meanwhile, compared with the pre-COVID-19 period, the ESG rating had a stronger impact on the NCSP during the post-COVID-19 period.

Our findings have important implications for company management and policy makers. Company managers should aim to enhance their ESG performance and increase their NCSP by improving their sustainability. Companies should incorporate ESG concepts into their operations to improve their performance and to allow them to better regulate their behavior. The past belief that maintaining a green environment, taking social responsibility, and improving corporate governance will only result in higher opportunity costs should be abandoned. It is necessary for companies to improve the quality of their own assets, achieve internal growth, and focus on the value created in the long term in order to boost their competitiveness in the market. Additionally, companies should make ESG information disclosures proactively to establish a good corporate image and gain more attention and support from investors. In order to improve their ESG performance, companies should aim to improve their accounting conservatism by considering their actual situation. It may be

necessary for companies to mitigate their agency conflicts, solve any financing problems, improve their surplus quality, and enhance their attractiveness to investors.

In recent years, China's ESG rating system has lacked unified rules, resulting in difficulties in comparing the ratings between different companies and between the same companies now and in the past. Rating agencies should aim to effectively capture the requirements of companies' carbon peak and carbon neutral strategies according to the actual situation in China, build a unified ESG evaluation standard, create an evaluation system for characteristics specific to China, and guarantee the comparability of the ESG evaluation results. Relevant governmental sectors should further improve the existing ESG performance-related policies and the ESG information disclosure system. The government should actively guide the ESG performance of companies and introduce specific unified ESG information disclosure requirements and index systems. It should also consider the principle of accounting conservatism and strictly require companies to disclose their accounting information in a timely, prudent, and detailed manner. Regulatory authorities should improve the regulatory system and the legal environment and clarify the related incentives and penalties. Meanwhile, companies should be encouraged to actively fulfill their environmental and social responsibilities and improve their corporate governance.

**Author Contributions:** Conceptualization, G.W. and A.Y.D.; methodology, G.W.; software, G.W.; validation, G.W. and A.Y.D.; formal analysis, G.W.; investigation, G.W.; resources, G.W.; data curation, G.W.; original draft preparation, G.W. and A.Y.D.; review and editing, G.W. and A.Y.D.; visualization, G.W. and A.Y.D.; supervision, A.Y.D.; funding acquisition, G.W. All authors have read and agreed to the published version of the manuscript.

**Funding:** This research received no external funding.

**Institutional Review Board Statement:** Not applicable.

**Informed Consent Statement:** Informed consent was obtained from all the subjects involved in this study.

**Data Availability Statement:** We used private sources of data; consistent with the contracts entered into by our academic institutions, we are not licensed to distribute such information. We refer readers to consult the sources cited and publicly available information.

**Acknowledgments:** We acknowledge the use of SynTao Green Finance data, available at the SynTaogy webpage (https://www.syntaogf.com/) (accessed on 20 March 2022).

**Conflicts of Interest:** The authors declare no conflict of interest.

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
