# Peer review of "ESG Rating and Northbound Capital Shareholding Preferences: Evidence from China"

_sustainability, doi:10.3390/su14159152_

Round 1

Reviewer 1 Report

My comments related to this paper are the following:

- in terms of literature review, in my opinion, the authors should present also other methods used to see how the variables are correlated, the impact would be interesting to find out;

- to detail more the way the authors have collected data (annually, quarterly, etc; consolidated financial statements, etc.), present how the companies selected are split into industries/sectors of activity and based on what type of classification;

- to detail why the ESG was selected as a Rating and not as a Score, because there is also this option;

- Some variables were logarithmic, please explain how you interpret data after the use of logarithm and the reason for doing it;

- the conclusion should put more accent on the novelty elements; if or if not the hypotheses were tested and the conclusions;

Author Response

Dear reviewer,

Best regards.

Reviewer 2 Report

The motivation of the study is weak or non-existent.

Although literature review section is given but Theoretical framework is also missing.

The author need to justify the sample i.e., years, number of firms, why only balanced panel, whats the total number of firm and whats the filtration criteria.

Report of results in not up to the mark. 

The results are reported but their economic meaning and significance is missing.

Practical implication of the study need to be highlighted. 

Abstract of the study didn't cover the study which need to be improved.

Author Response

Dear reviewer,

Best regards.

Reviewer 3 Report

Overall, this is a very interesting paper that studies the relationship between ESG ratings and institutional investor preferences in China. The paper is methodologically and quantitatively sound and is original in its main contribution, showing that firms with higher ESG ratings enjoy higher levels of ownership from "Northbound capital" share-connect programs. The findings with regards to governance being the most important ESG dimension as well as quite inventive and rigorous endogeneity tests are also very interesting. However, I believe there is some room for improvement, mostly with regards to synthesis of findings with prior literature and somewhat with regards to estimations. Next, my comments are outlined from most to least important.

1) Table 11, for instance, presents regression results with E, S, and G components separately. Why not regress on all three components simultaneously? As these can be quite tightly correlated, only this specification can truly show which of the three dimensions is the most important for northbound capital shareholding. 

2) As the paper argues northbound capital shareholding represents "smart money" and "hot money", it would be interesting to link this study's findings to the existing literature that investigates ESG ratings and their effects on stock returns from an institutional investment perspective (e.g., Cao et al., 2020; Shanaev and Ghimire, 2022).

3) The estimation strategy of the study mainly revolves around calculating treatment effects with regards to northbound capital shareholding and ESG ratings. As the final year of the sample is 2020, these can be affected by the impact COVID-19 had on ESG investor preferences which is highlighted in the literature (e.g., Broadstock et al., 2021, which this paper already cites, or Feng et al., 2022). A more detailed discussion or at least an acknowledgement of this would be welcome. 

4) There are some awkward word and phrase choices throughout the manuscript. Please consider proofreading.

References:

Broadstock, D., Chan, K., Cheng, L., & Wang, X. (2021). The role of ESG performance during times of financial crisis: Evidence from COVID-19 in China. Finance Research Letters, 38, 101716.

Cao, J., Titman, S., Zhan, X., & Zhang, W. (2020). ESG preference, institutional trading, and stock return patterns (No. w28156). National Bureau of Economic Research.

Feng, J., Goodell, J., & Shen, D. (2022). ESG rating and stock price crash risk: Evidence from China. Finance Research Letters, 46, 102476.

Shanaev, S., & Ghimire, B. (2022). When ESG meets AAA: The effect of ESG rating changes on stock returns. Finance Research Letters, 46, 102302.

Author Response

Dear reviewer,

Best regards.

Reviewer 4 Report

With regard to Table 4, was a Hausman test applied? It seems that it was not because both the Fixed effects and Random Effects are presented. Only one is appropriate.

The paper needs careful editing. Run-on sentences, etc., must be corrected.

Author Response

Dear reviewer,

Best regards.

Round 2

Reviewer 1 Report

Comments after major revision:

I did not find the reason why the authors used ESG rating rather than ESG score, why they have not took into consideration ESG controversies;

How data is interpreted after using the logit model.

The novelty of the research.

Reviewer 2 Report

All the suggested changes have been incorporated. Don't have any further comment. Best of luck with the manuscript.

Reviewer 4 Report

As I read this paper, the most obvious characteristic is its lack of clarity. This begins in the Abstract and is found throughout the paper. It is tedious reading.

P. 1.: the post-COVID 19 period--how long is it, precisely. 

P. 1. "...than it did before."  Before what?

p2. "...before." Before what. 

P. 3. "Chapter". What are you talking about?

P.4. item (3). This needs elaboration. 

P.7 at bottom. Problem: there is no calculation.

P. 8 Explain "Post" and "Treat" more fully

P. 9. Explain how you feel it is OKAY to use  an interaction term as the Dependent variable.

Every sentence for clarity.

Round 3

Reviewer 4 Report

Much improved.